# Noise Level and Similarity Analysis for Computed Tomographic Thoracic Image with Fast Non-Local Means Denoising Algorithm

**Bae-Guen Kim** [1], **Seong-Hyeon Kang** [1], **Chan Rok Park** [2], **Hyun-Woo Jeong** [3,*] **and Youngjin Lee** [1,*]

[1] Department of Radiological Science, Gachon University, 191, Hambakmoero, Yeonsu-gu, Incheon 21936, Korea; kbg1229@gc.gachon.ac.kr (B.-G.K.); tjdgus7345@gc.gachon.ac.kr (S.-H.K.)

[2] Department of Radiological Science, Jeonju University, 303, Cheonjam-ro, Wansan-gu, Jeonju-si 55069, Korea; tigeaglepcr@jj.ac.kr

[3] Department of Biomedical Engineering, Eulji University, 553, Sanseong-daero, Sujeong-gu, Seongnam-si, Gyeonggi-do 13135, Korea

* Correspondence: hwjeong@eulji.ac.kr (H.-W.J.); yj20@gachon.ac.kr (Y.L.);
Tel.: +82-31-740-7135 (H.-W.J.); +82-32-820-4362 (Y.L.)

**Abstract:** Although conventional denoising filters have been developed for noise reduction from digital images, these filters simultaneously cause blurring in the images. To address this problem, we proposed the fast non-local means (FNLM) denoising algorithm which would preserve the edge information of objects better than conventional denoising filters. In this study, we obtained thoracic computed tomography (CT) images from a male adult mesh (MASH) phantom modeled by computer and a five-year-old phantom to perform both the simulation study and the practical study. Subsequently, the FNLM denoising algorithm and conventional denoising filters, such as the Gaussian, median, and Wiener filters, were applied to the MASH phantom image adding Gaussian noise with a standard deviation of 0.002 and practical CT images. Finally, the results were compared quantitatively in terms of the coefficient of variation (COV), contrast-to-noise ratio (CNR), peak signal-to-noise ratio (PSNR), and correlation coefficient (CC). The results showed that the FNLM denoising algorithm was more efficient than the conventional denoising filters. In conclusion, through the simulation study and the practical study, this study demonstrated the feasibility of the FNLM denoising algorithm for noise reduction from thoracic CT images.

**Keywords:** noise analysis; computed tomography thoracic image; fast non-local means approach; denoising algorithm; MASH phantom

## 1. Introduction

Recently, the speed, accuracy, and convenience of identifying the type and location of a disease have improved using diagnostic imaging modalities such as magnetic resonance imaging (MRI), positron emission tomography (PET), single photon emission computed tomography (SPECT), and computed tomography (CT), which is the most commonly used modality [1,2].

However, noise is inevitably caused by scattering rays, quantum noise, and detectors, as the images for medical diagnosis are displayed as digitized images through the conversion process of each device. Such noise that degrades the diagnostic accuracy is fatal for patients. In particular, noise is commonly observed under low-dose examinations and thoracic CT images are usually obtained under low-dose examinations to obtain improved contrast [3,4].



In the case of lung cancer, which is the most common cause of cancer death, various studies have demonstrated the feasibility of applying low-dose examinations for early diagnosis of lung cancer. A study by the National Lung Screening Trial Research Team showed the usefulness of low-dose CT [5]. In addition, the use of low-dose examinations is on the rise with the increase in awareness of the risk of ionizing radiation [6]. However, noise is inevitably generated due to the use of low-dose examinations [7–9].

The reduction of noise as much as possible is necessary to avoid misdiagnosis by doctors. Therefore, to reduce noise, denoising filters such as Gaussian, median, and Wiener filters have been developed. In a Gaussian filter, after setting up the rotating symmetrical mask, the central pixel is calculated by increasing the weight as the standard deviation of the Gaussian distribution increases [10]. A median filter replaces its median masks with the median values of the surrounding pixels. The Wiener filter reduces noise by separating the image signal from the noise in the frequency domain. However, conventional denoising filters are applied equally to the entire image. Therefore, these are inefficient in filtering nonlinear noise and cause signal distortion. In particular, the image sharpness decreases due to the blurring effect, which is caused by an excessive thinning of the outline [11].

To solve the above-mentioned problems, the non-local means (NLM) denoising algorithm was developed. The NLM denoising algorithm equalizes the interior pixel values while maintaining the outline by setting regions of interest (ROIs) and calculating the weight based on the intensities of the adjacent pixels and Euclidean distance. Therefore, the NLM denoising algorithm can offset the blurring effect [12]. Naegelad et al. specifically applied the NLM denoising algorithm to MRI. They showed that the NLM algorithm preserved nearly 100% of the edge information in the image, while achieving the best signal-to-noise ratio (SNR) as compared with conventional denoising algorithms. Although they suggestedthe use of the NLM denoising algorithm in the clinical field on the basis of their research, there are time constraints due to the computational complexity [13]. To address this problem, we developed the fast non-local means (FNLM) denoising algorithm, which reduced the computational complexity by changing the weight calculations of the NLM denoising algorithm from two dimensions to one dimension. This technique can improve the time resolution of the NLM denoising algorithm [14–18].

In particular, this study was performed in both simulation study and practical study using a male adult mesh (MASH) phantom and a five-year-old phantom. Therefore, the radiation exposure was prevented and more practical results were derived by applying FNLM denoising algorithm to actual CT image.

Here, we demonstrated the feasibility of the FNLM denoising algorithm for noise reduction from thoracic CT images.

## 2. Materials and Methods

### 2.1. Acquisition of the Thoracic Image from the Male Adult Mesh (MASH) Phantom: The Simulation Study

The MASH phantom is a computational model of human anatomy for research in medicine and radiology. These phantoms are realized using the polygon mesh technique, which uses a collection of vertices, edges, and faces to define the shape of a polyhedral object with three-dimensional (3D) computer graphics. The MASH phantom is more similar to the human body than the conventional medical internal radiation dose (MIRD5) phantom. The advantages of using the MASH phantom are that variables corresponding to changes in the posture of the human body and irradiation conditions can be controlled, along with preventing patient/subject exposure. The MASH phantom was designed to represent the standing position of a male adult with 113 organs, bones, and tissues according to the recommendations of report 89 of the ICRP. The pixelized phantom shows 97.4% consistency with the ICRP data [19–21].

This image processing was performed using MATLAB (version R2020a, The MathWorks, Inc., Natick, MA, USA). An axial thoracic image was obtained from the CT scan dataset of the MASH phantom. Then, Gaussian noise with a standard deviation of 0.002 was added to the image. Figure 1

shows the original thoracic image with the target and background ROIs. The region within the blue box was magnified to highlight the reconstruction results. ROI$_{target}$ and ROI$_{background}$ were set over the cardiac and lung regions, respectively.

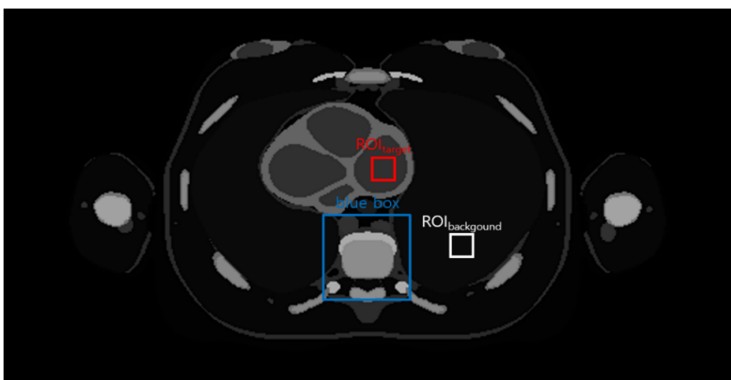

**Figure 1.** The image obtained from the male adult mesh (MASH) phantom. The region inside the blue box was magnified to compare the denoised images. Regions of interests (ROIs) were set to calculate the coefficient of variation (COV) and contrast to noise ratio (CNR) of the obtained chest image.

### 2.2. Acquisition of the Thoracic Image from the Five-Year-Old Phantom: The Practical Study

A 5-year-old phantom (Model ATOM 704, CIRS Inc., Norfolk, VA, USA) was scanned using a 64-detector row scanner (SOMATOM Definition AS+, Siemens, Malvern, PA, USA). The CT examination parameters were set as follows: tube potential, 120 kVp; tube current, 23 mA; exposure time, 300 ms; and slice thickness, 1 mm. Then, the axial image of $512 \times 512$ size was obtained. Generally, the low-dose CT examinations are performed with 30–50 mAs. However, in this study, the parameters of 7 mAs were set in order to show that the FNLM denoising algorithm was efficient at low exposure conditions. Moreover, in modern CT examination, the automatic current technique has been utilized to compensate the differences of thickness of body. However, in this examination, the fixed current was utilized to show the clear results by excluding some random parameters. In this experiment, we did not add the Gaussian noise to the image that we obtained to derive practical results from an unaffected condition. Figure 2 shows the initiative image of the 5-year-old phantom. The ROIs and blue box in the image were set as the similar regions to the image of the MASH phantom.

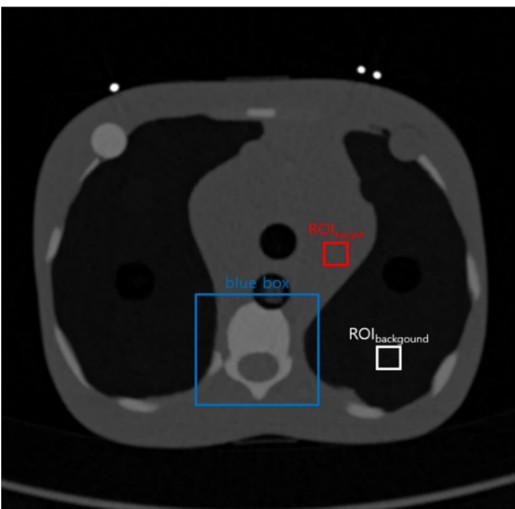

**Figure 2.** The image obtained from a 5-year-old phantom. The region inside the blue box was magnified to compare the denoised images. ROIs were set to calculate the coefficient of variation (COV) and contrast-to-noise ratio (CNR) of the chest image.

### 2.3. Fast Non-Local Means Algorithm

The NLM denoising algorithm is effective for noise reduction. In addition, this algorithm can resolve the problems of conventional denoising filters, such as signal distortion and blurring. This is because the NLM denoising algorithm compares the overall geometric composition using the Euclidean distance as a weight as opposed to using the sliding method on the entire image, such as in the case of the Gaussian, median, and Wiener filters. The NLM denoising algorithm can be defined as follows:

$$NL[I](m) = \sum_{N \in I} \omega(m, n) I(n), \tag{1}$$

where the weight $\omega(m, n)$ is defined as follows:

$$\omega(m, n) = \frac{1}{Z(m)} \sum e^{-\frac{G_\sigma(\tau) \|I(M+\tau) - I(n+\tau)\|_2^2}{d^2}}, \tag{2}$$

where $\tau$ is the number of pixels ; $G_\sigma(\tau)$ is the Gaussian distribution with size $\sigma^2$ of the number of background pixels; $\|I(M + \tau) - I(n + \tau)\|_2^2$ is the intensity difference between adjacent pixels based on the Euclidean distance values; and $Z(m)$ sets the leveling constant, as follows:

$$Z(m) = \sum_n e^{-\frac{G_\sigma(\tau) \|I(m+\tau) - I(n+\tau)\|_2^2}{d^2}}. \tag{3}$$

The FNLM denoising algorithm modifies the calculation of $\omega(m, n)$ in the NLM denoising algorithm from one dimension to two dimensions. The modified $\omega(m, n)$ is defined as follows:

$$\omega(m, n) = \frac{1}{Z(m)} H_i(I(m + s) - I(m - s)), \tag{4}$$

where $\tau$ is defined as $n - m$, s is defined as $m + \tau$, and $H_i$ is defined as follows:

$$H_i(s) = \sum_{q=0}^{s} e^{-\frac{\|I(q) - I(q+\tau)\|_2^2}{d^2}}. \tag{5}$$

Compared with the NLM denoising algorithm, the FNLM denoising algorithm theoretically improves the time resolution by approximately 4× by simplifying the process through one-dimensional computations [12–18,22].

### 2.4. Quantitative Evaluation Factors

#### 2.4.1. Evaluation Factors for the Noise Level

In this study, noise leveling factors such as the coefficient of variation (COV) and contrast-to-noise ratio (CNR) were employed [23,24]. ROIs were set to calculate the COV (using the target ROI) and CNR (using the target and background ROIs). The COV is an evaluation factor that quantitatively represents the noise in an image. A superior image has a low COV. In addition, the CNR is a factor that evaluates the contrast and noise of images in combination. A superior image has a high CNR. The COV and CNR are defined as follows:

$$COV = \frac{\sigma}{\mu}, \tag{6}$$

$$CNR = \frac{|S_A - S_B|}{\sqrt{\frac{1}{2}\left(\sigma_A^2 + \sigma_B^2\right)}}, \tag{7}$$

where $\sigma$ is the standard deviation of the signal ; $\mu$ is the mean value of the signal ; $S_A$ and $\sigma_A$ are the mean and standard deviation of the signal in the target ROI, respectively; and $S_B$ and $\sigma_B$ are the average and standard deviation of the background signal, respectively.

### 2.4.2. Evaluation Factors for Similarity

The peak signal-to-noise ratio (PSNR) and correlation coefficient (CC) were used to evaluate the similarity [25,26]. A superior image has a high PSNR, which is primarily used to evaluate the loss of image quality. However, the PSNR of lossless image is undefined because its mean squared error (MSE) is 0. In addition, CC represents the relationship between the original and denoised images and assumes values in the range of $-1$ to $+1$, where $\pm1$ indicates the strongest possible concordance and 0 the strongest possible discordance. The CC is equal to the ratio of the covariance and the product of the standard deviations of original and denoised images. The PSNR and CC are defined as follow:

$$PSNR = 10 \times \log\left(\frac{MAX_I^2}{MSE}\right),\tag{8}$$

$$CC = \frac{\sum_i^n\left(X_i - \overline{X}\right)\left(Y_i - \overline{Y}\right)}{\sqrt{\sum_i^n\left(X_i - \overline{X}\right)^2}\sqrt{\sum_i^n\left(Y_i - \overline{Y}\right)^2}},\tag{9}$$

where $MAX_I$ in Equation (8) is the maximum signal value and $MSE$ is defined as follow:

$$MSE = \frac{1}{M \times N}\sum_{i=1}^{N}\sum_{j=1}^{M}\left[I(i,j) - I_n(i,j)\right]^2,\tag{10}$$

where $I$ is a grayscale image with a size of $M \times N$ size and $I_n$ is the noisy image.

## 3. Results

### 3.1. Results from MASH Phantom

Figure 3 shows the region inside the blue box that was magnified to clearly illustrate the performance of each noise reduction algorithm. The original (top left, Figure 3), noisy (top right, Figure 3), Gaussian (center left, Figure 3), median (center right, Figure 3), Wiener (bottom left, Figure 3), and FNLM (bottom right, Figure 3) images are depicted. We performed qualitative visual evaluation of the images obtained by applying the conventional denoising filters and FNLM denoising algorithm. Subsequently, the performance of the FNLM denoising algorithm was evaluated in terms of various quantitative evaluation factors.

Figure 4 shows the COV and CNR results for evaluating the noise level of the reconstructed images. In the noise level evaluation, the FNLM denoising algorithm achieved the best performance as compared with the conventional denoising filters. The COV values of images applied the noisy, Gaussian, median, Wiener, and proposed FNLM denoising algorithm were 0.237, 0.154, 0.087, 0.046, and 0.023, respectively. The COV of the image obtained with the FNLM denoising algorithm was approximately 10.3×, 6.7×, 3.8×, and 2.02× higher than that of the noisy, Gaussian, median, and Wiener images, respectively. Moreover, the CNRs of the noisy, Gaussian, median, Wiener, and proposed FNLM denoising algorithm were 4.908, 7.600, 11.947, 25.511, and 55.627, respectively. The CNR of the image obtained with the FNLM denoising algorithm was approximately 11.33×, 7.32×, 4.66×, and 2.18× higher than that of the noisy, Gaussian, median, and Wiener images, respectively.

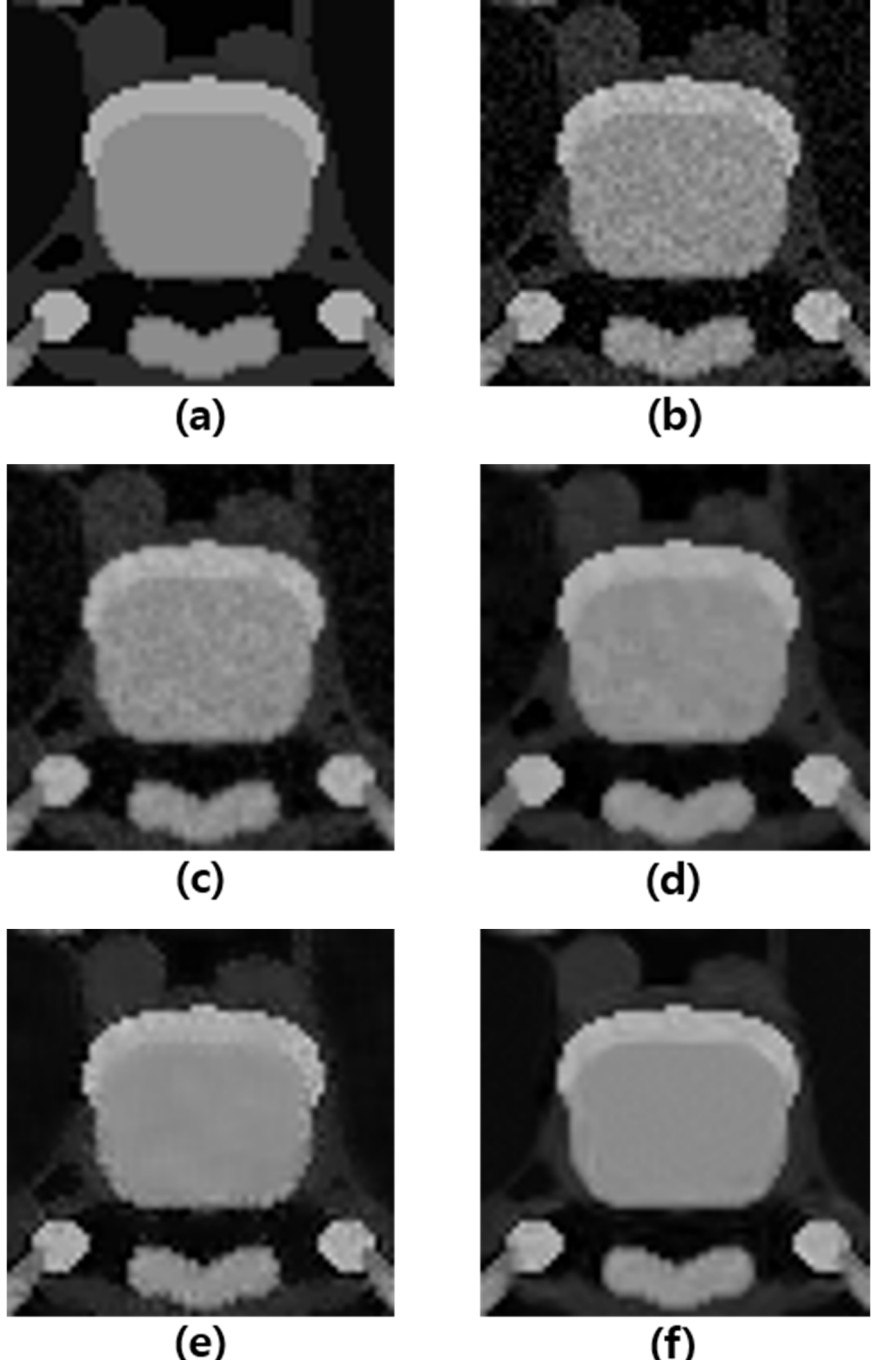

**Figure 3.** Magnified views (region inside the blue box). (**a**) Original; (**b**) Noisy; (**c**) Gaussian filtered; (**d**) Median filtered; (**e**) Wiener filtered; (**f**) Fast non-local means (FNLM) filtered images.

Figure 5 shows the PSNR and CC results for evaluating the similarity between the original and reconstructed images. In the similarity evaluation, the FNLM denoising algorithm again showed the most efficient value as compared with the conventional denoising filters. The PSNRs of the noisy, Gaussian, median, Wiener, and proposed FNLM denoising algorithm were 77.261, 79.537, 82.094, 81.882, and 82.354, respectively. The FNLM denoising algorithm had the highest PSNR, followed by the median, Wiener, Gaussian, and noisy images. Moreover, the CC values of the noisy, Gaussian, median, Wiener, and proposed FNLM denoising algorithm were 0.958, 0.977, 0.985, 0.990, and 0.990, respectively. Unlike the PSNR results, the FNLM denoising algorithm had the highest CC value, followed by the Wiener, median, Gaussian, and noisy images.

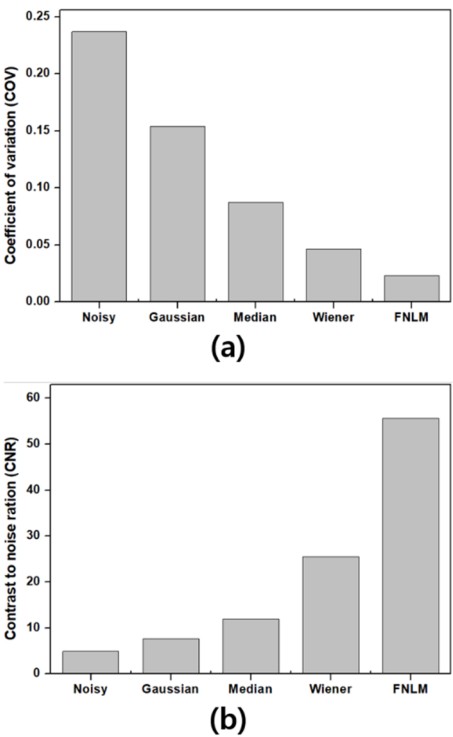

**Figure 4.** The (**a**) coefficient of variation (COV) and (**b**) contrast-to-noise ratio (CNR) metrics computed in the set ROIs to evaluate the noise level of the reconstructed image obtained from the MASH phantom.

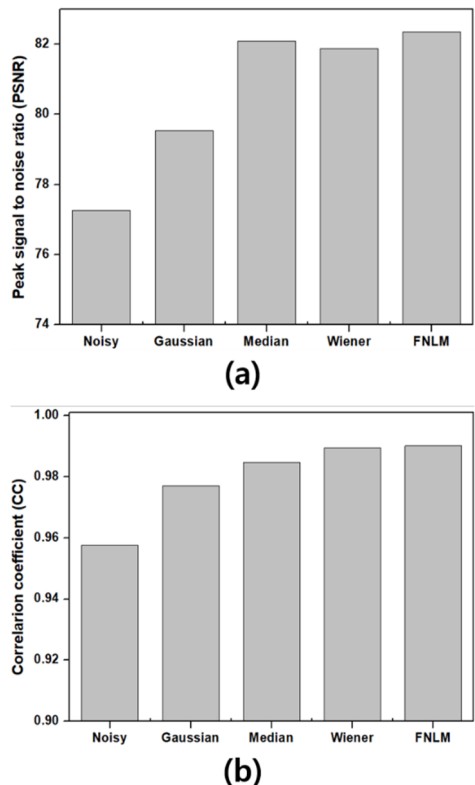

**Figure 5.** The (**a**) peak signal-to-noise ratio (PSNR) and (**b**) correlation coefficient (CC) values used to evaluate the similarity between the reconstructed images and original image.

### 3.2. Results from the Five-Year-Old Phantom

Figure 6 shows the region inside the blue box that was magnified to clearly illustrate the performance of each noise reduction algorithm. The original (top left, Figure 6a), Gaussian (top center, Figure 6a), median (top right, Figure 6c), Wiener (bottom left, Figure 6d), and FNLM (bottom right, Figure 6e) images are depicted. There was not any additional noise except for the natural noise of the image acquisition process. As a result of observing the images of Figure 6, the image in Figure 6e was the clearest. Subsequently, the performance of the FNLM denoising algorithm was evaluated in terms of the two noise level evaluation factors.

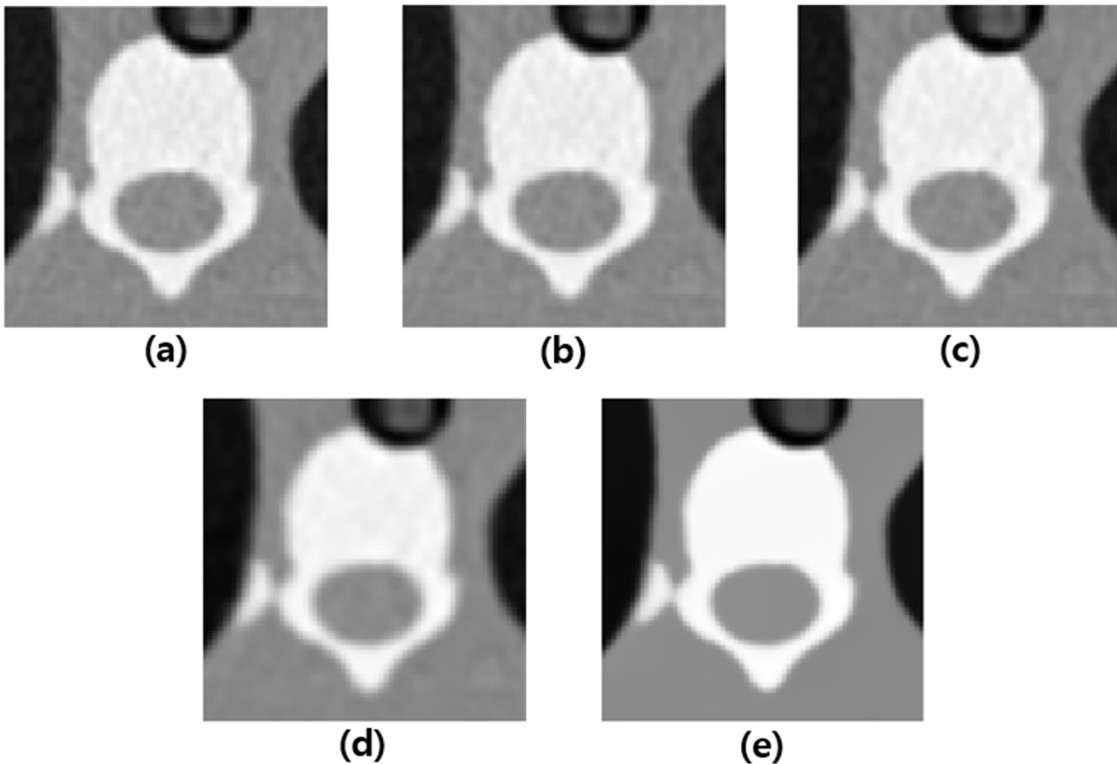

**Figure 6.** Magnified views (region inside the blue box). (**a**) Original; (**b**) Gaussian filtered; (**c**) Median filtered; (**d**) Wiener filtered; (**e**) FNLM filtered images.

Figure 7 shows the COV and CNR results for evaluating the noise level of the reconstructed images of the five-year-old phantom. In the noise level evaluation of these images, the FNLM denoising algorithm achieved the best performance as compared with the conventional denoising filters. The acquired COV values of images applied the original, Gaussian, median, Wiener, and proposed FNLM denoising algorithm were 0.000798, 0.000751, 0.000703, 0.000497, and 0.000102, respectively. The COVs of the images obtained with the FNLM denoising algorithm were approximately 7.82×, 7.36×, 6.89×, and 4.87× better than that of the original, Gaussian, median, and Wiener images, respectively. Moreover, the CNRs of the original, Gaussian, median, Wiener, and proposed FNLM denoising algorithm were 45.138, 47.630, 50.527, 67.125, and 236.635, respectively. The CNRs of the image obtained with the FNLM denoising algorithm were approximately 5.24×, 4.97×, 4.68×, and 3.53× higher than that of the original, Gaussian, median, and Wiener images, respectively.

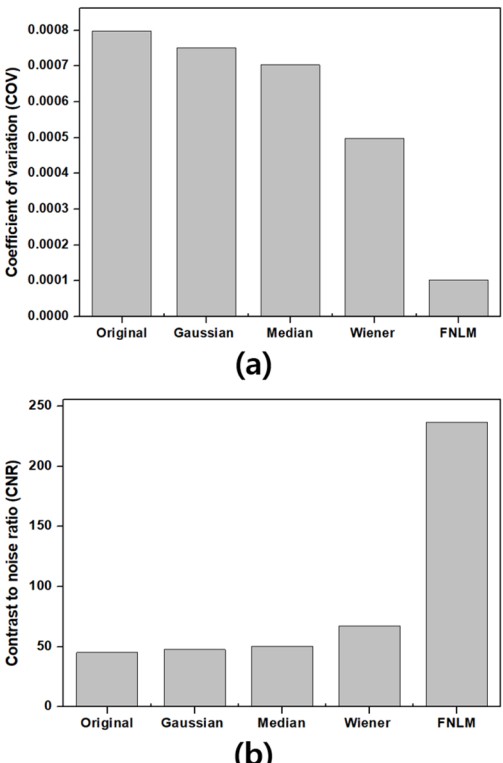

**Figure 7.** The (**a**) coefficient of variation (COV) and (**b**) contrast-to noise-ratio (CNR) metrics computed in the set ROIs to evaluate the noise level of the reconstructed image obtained from the 5-year-old phantom.

## 4. Discussion

Lung cancer consistently has the highest mortality rate among cancers. Therefore, early diagnosis through regular monitoring using CT is critical to decrease the death rate, which has also been proven to work in practice. Unfortunately, CT, which is based on X-rays, is accompanied by potential risks because of its high-dose exposure. Meanwhile, it is important to find the peripheral lung nodules as they can become cancerous. Recently, the feasibility of using low-dose examination for the discovery of peripheral lung nodules has been demonstrated and various studies have showen that low-dose examination can replace the conventional methods [27–29]. The exposure dose of the subjects can be significantly reduced by low-dose examination.

However, low-dose examination inevitably leads to a large amount of noise caused by scattering rays, leading to misdiagnosis in some cases. Accordingly, several algorithms have been developed to reduce the noise. However, conventional local denoising filters cause blurring of the edges in the images. To address this problem, a non-local algorithm was selected in this study.

The purpose of this study was to demonstrate the feasibility of the FNLM denoising algorithm for noise reduction from thoracic CT scans. In the first process of this experiment, axial thoracic images were obtained from CT data of a MASH phantom using the MATLAB program, and then Gaussian noise with a standard deviation of 0.002 was added to these axial thoracic images. The Gaussian-Poisson mixture is general noise in the images based on X-ray [30]. The reason why the Poisson was not considered in this study is that the Poisson noise is affected by the modalities. It is usually caused due to the lack of photons. Therefore, we considered the Gaussian noise that represents the normal distribution overall. The Gaussian, median, Wiener, and FNLM denoising algorithms were, then, applied to the noisy images. Subsequently, the filtering results were compared using quantitative evaluation factors. Here, COV and CNR were used to evaluate the degree of noise after application of the algorithms. Actually, there are a few methodologies for noise estimation [31]. In this study, the general method that calculates the variation of pixel values of the region that is composed of the same matter was utilized. The variation can be considered to be noise because the pixel values of the

same matter should be the same. Continuously, the PSNR was used to assess the amount of image quality loss, and the CC was used to evaluate the correlation between the images. Next, the image from the five-year-old phantom was obtained under practical conditions to verify whether the proposed algorithm could be used in the clinical field or not.

The process of selecting a phantom and the characteristics of lung cancer, such as gender and age, were not considered, because we focused on demonstrating the performances about noise reduction and edge information preservation. We supposed that according to the hypothesis associated with the relationship between the density and the attenuation, this phantom that has similar composition with human anatomy can derive useful results [32,33]. However, for more reliable results, utilizing a phantom that is similar to the MASH phantom is ideal. The process of this experiment is the same as the process of the first. However, we did not add the Gaussian noise and evaluated the noise level only. As shown by the results, the proposed FNLM denoising algorithm showed the best performance in every evaluation regardless of the virtual and practical image.

Although, originally, the PSNR was optimized to evaluate the quality of the compressed image, in this study, the PSNR was used to evaluate the degree of image restoration achieved by various denoising algorithms. The PSNR and CC values can differ from visual evaluation, as these factors are measured mathematically. However, as shown in Figures 3 and 6, the image that was the applied FNLM denoising algorithm visually showed the clearest image without any blurring. To prove the enhancement of noise level and sharpness, we measured the signal intensity of the images. Figures 8 and 9 show the signal intensity profiles of the images that are the applied Wiener filter and FNLM denoising algorithm, respectively. The Wiener filter, only, was analyzed to compare with the FNLM denoising algorithm because the Wiener filter showed the best performance in almost all of the evaluations among the conventional denoising filters.

Figure 8 is the signal intensity profile of the image that was obtained from the MASH phantom. The spine region was utilized to obtain this profile. The boxes represent the noise level in the region that should represent smooth signal intensity. The gap of signal intensity between the highest and the lowest in box_a is larger than the gap of signal intensity in box_b. In other words, it means that box_a represented the mottled image and the circles represented the sharpness of the edge, circle_b showed the definite contrast as compared with circle_a, and moreover, the edge information on the opposite of circle_a was distorted. On the contrary, the profile in Figure 8b represents symmetrical result because the FNLM denoising algorithm considers the geometrical composition of the entire image by using the Euclidean distance.

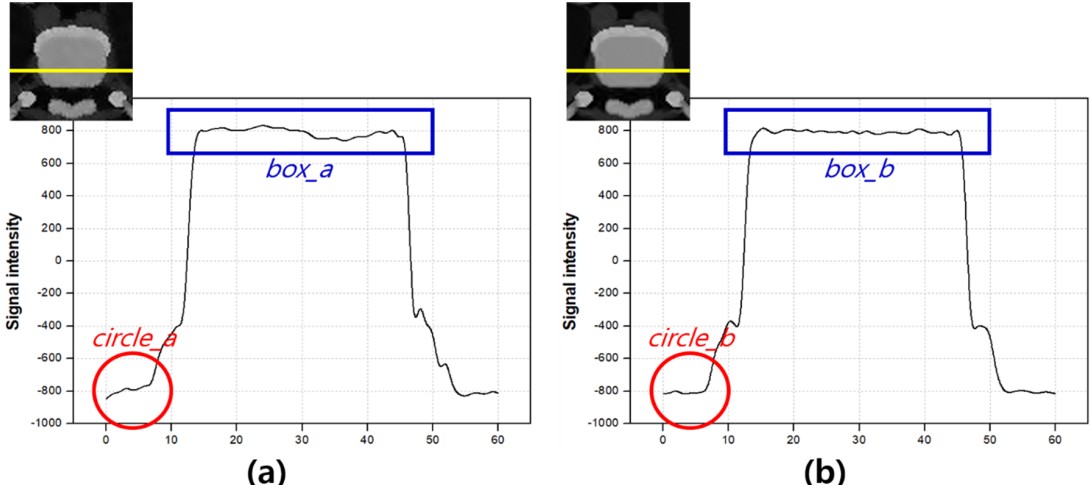

**Figure 8.** The (**a**,**b**) are signal intensity profiles of the MASH phantom images applied the Wiener filter and the FNLM denoising algorithm, respectively. The graphs represent the signal intensity of the yellow lines in the images. The boxes represent the denoising performance of the Wiener filter and the FNLM denoising algorithm. The circles represent the sharpness of the images.

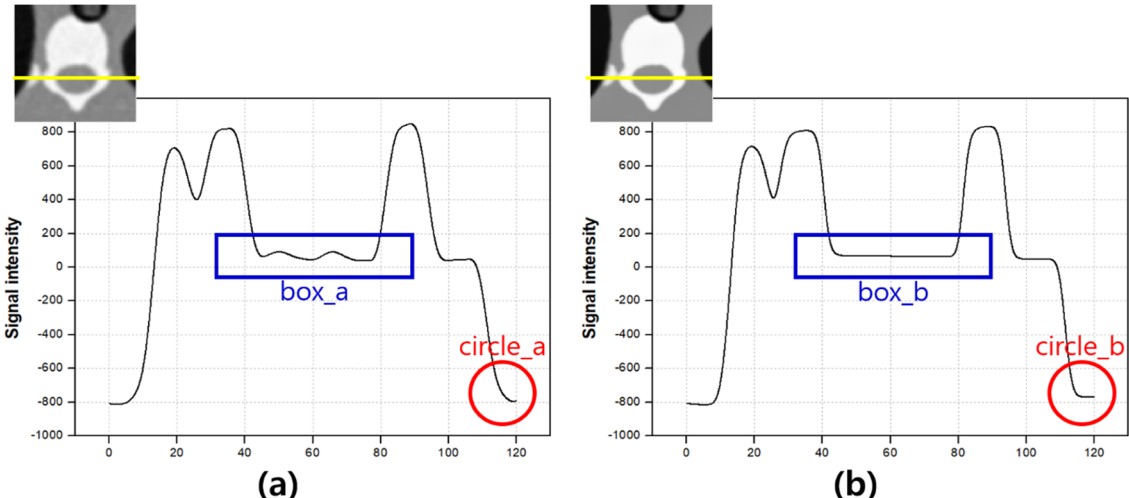

**Figure 9.** (**a**,**b**) are signal intensity profiles of the 5-year-old phantom images applied the Wiener filter and the FNLM denoising algorithm, respectively. The graphs represent the signal intensity of the yellow lines in the images. The boxes represent the denoising performance of the Wiener filter and the FNLM denoising algorithm. The circles represent the sharpness of the images.

Figure 9 is the signal intensity profiles of the images that were obtained from the five-year-old phantom. The spine region was utilized similar to the image of the MASH phantom. The boxes also represent the noise level in the region that should represent smooth signal intensity. In the profile in Figure 9b, the signal intensity of box_b is almost flat and also the contrast of circle_b definitely show as compared with the profile in Figure 9a. Therefore, we confirmed that the FNLM denoising algorithm is more efficient than the conventional denoising filters for reducing the noise and improving sharpness by measuring the signal intensity of each image.

In addition, the actual computational times were measured using a MATLAB program to compare the time resolution of the FNLM and NLM denoising algorithms. The actual operation time was measured and the average of 20 measurements was calculated. As a result, the average computational time of the FNLM denoising algorithm was 0.998 s, while that of the NLM denoising algorithm was 17.428 s. The time resolution of the FNLM denoising algorithm was approximately 17.5× better than the conventional algorithm due to a simplified computation of its weights. Observing Equation (2), the X-axis ($m$) and Y-axis ($n$) components of the image are defined in this equation. In other words, the presence of two axes necessitates complicated computations along two dimensions. Therefore, as shown in Equation (5), the size of the matrix was reshaped from $(m, n)$ to $(m \times n, 1)$ because the computer treats $(m \times n, 1)$ as a one-dimensional vector, thus, simplifying the computational complexity [34,35]. Figure 10 shows the time resolution of the FNLM and NLM algorithms. It can be observed that the proposed FNLM denoising algorithm significantly improved the time resolution as compared with the conventional NLM denoising algorithm.

In this study, initially, we verified that the noise level and similarity of the thoracic CT image obtained from the MASH phantom are most efficiently improved by the FNLM denoising algorithm as compared with the conventional denoising filters, and then we applied the FNLM denoising algorithm to the practical thoracic CT image to demonstrate the feasibility of the proposed algorithm in the clinical field. In addition, we measured the time resolution to compare the FNLM and NLM denoising algorithms. Thus, it was demonstrated that computations could be effectively simplified by vectorization.

Consequently, we proposed a solution to address the disadvantages of low-dose examination and conventional denoising filters. Thus, we expect that the application of the FNLM denoising algorithm can contribute to the acquisition of high-quality images for the early diagnosis of lung cancer with as little exposure as possible.

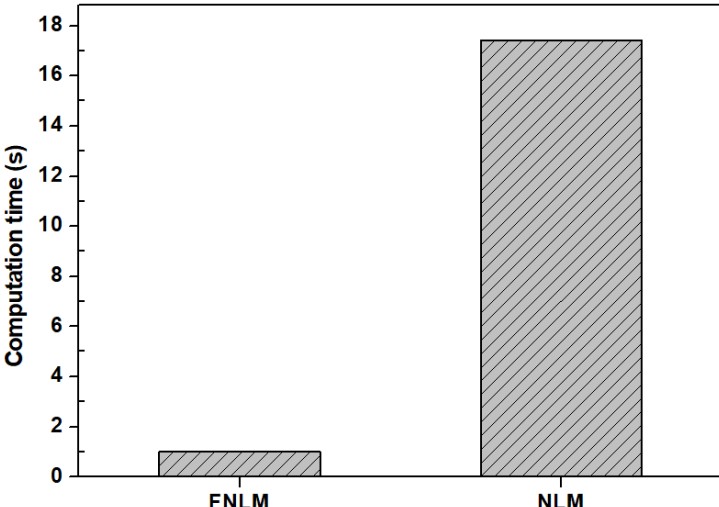

**Figure 10.** Comparison between the time resolutions of the FNLM and non-local means (NLM) denoising algorithms.

However, the results reported in this study may vary according to some conditions of the utilized images. According to the study of Korn et al., because the reconstruction algorithms affected the quality of the image, the results reported in this study coud be affected when the different reconstruction algorithms were applied to the image [36]. Then, the results may vary depending on whether artifacts such as star-like artifact, beam hardening artifact, and ringing artifact, are present or not in the image. Because the artifacts affect the geometric structure of image, the FNLM denoising algorithm based on the Euclidean distance may show varied results.

## 5. Conclusions

By comparing the images obtained with the FNLM denoising algorithm and other conventional denoising filters, we demonstrated that the FNLM denoising algorithm achieved the best denoising performance and similarity with the original image. In addition, we demonstrated that the FNLM denoising algorithm improved the time resolution significantly as compared with the NLM denoising algorithm.

In conclusion, we expect that the FNLM denoising algorithm can contribute to reducing the ratio of misdiagnosis of thoracic disease and the exposure dose because the noise, which is the advantage of the low-dose examination, can be suppressed efficiently by the FNLM denoising algorithm. Future studies will focus on further improving the proposed algorithm to elicit better image characteristics.

**Author Contributions:** Conceptualization, B.-G.K., H.-W.J., and Y.L.; Data curation, B.-G.K., S.-H.K., and C.R.P.; Formal analysis, B.-G.K., S.-H.K., and Y.L.; Funding acquisition, H.-W.L.; Software, S.-H.K. and Y.L.; Writing—original draft, B.-G.K., S.-H.K., and C.R.P.; Writing—review and editing, H.-W.L. and Y.L. All authors have read and agreed to the published version of the manuscript.

**Funding:** This work was supported by the Technology Development Program (S2827843) funded by the Ministry of SMEs and Startups (MSS, Korea).

**Conflicts of Interest:** The authors declare no conflict of interest.

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
