# Peer review of "Noise Level and Similarity Analysis for Computed Tomographic Thoracic Image with Fast Non-Local Means Denoising Algorithm"

_applsci, doi:10.3390/app10217455_

Round 1

Reviewer 1 Report

Good paper, about a very interesting matter. However, I think the work can be improved, according to the following suggestions:

  • In the text I note that all measurement data, used instrumentation whith their uncertainty characteristics, and used algorithm(s) are quite missing, then a reader cannot verify the correctness of calculations and of final sentences.
  •  
  • I think that the analysis reported in the work is relevant only to the case and to the phantom types studied. Can the Authors illustrate if and how it is possible to extend their proposed tecnique to all possible case-studies, so to generalize the procedure?
  • I suggest to increase the References List to reach a larger citation basis, by adding some articles about the treated matter and/or about correlated ones, as:

    1) The Journal of Polish Society of Medical Physics, Volume 26 (2), June 2020, ISSN 1898-0309, doi: 10.2478/pjmpe-2020-0013

    2) IEEE Transactions on Instrumentation and Measurement, Volume 66 (10), October 2017, pp. 2535-2544 

    3) J. of Thoracic and Cardiovascular Surgery November 2002, pp.1014-1020

    4) IEEE Transactions on Instrumentation and Measurement, Volume 63 (5), May 2014, pp. 1163-1170

    5) Proceedings of SPIE - The International Society for Optical Engineering, Vol. 6919, 69190I, March 2008, DOI: 10.1117/12.771009
  •  
  • 6) Respiration (2018); Vol. 96; pp. 231–239, DOI: 10.1159/000489177

  •  

Author Response

Thank you for review and comment in this manuscript.
We have revised the paper as your suggestion and responded point by point.
Please confirm attached revised manuscript and response files.

Best regards,

Youngjin Lee

Reviewer 2 Report

Manuscript applsci-949078

This manuscript reported how different denoising algorithms can affect the quality of CT images in chest studies. Thank you for giving me the opportunity of reviewing this manuscript. The topic is interesting and the paper is free of major grammar or spelling errors. However, it is my opinion that the study needs to be improved before publication. I have several major and minor comments listed below.

Major comments

  1. There are several methods to estimate noise in radiology (A. Malkus, T. P. Szczykutowicz, A method to extract image noise level from patient images in CT, Medical Physics, 2017, https://doi.org/10.1002/mp.12240). How did the authors decide to follow the reported methodology? Clarification is needed.
  2. It is not clear the reason why MASH and ATOM 704 phantoms were used in the study. The authors correctly stated that lung cancer is one of the most common cause of cancer death (some references would be helpful), but there is not any comments about the age stratification of such cancer death risk. This is critical because the entire study was designed using an adult computational phantom and a pediatric 5-years old physical phantom. Why did authors decide to utilize those phantoms? Why one computational and one physical? Why 5-year old? What about the difference in gender and age correlated to cancer risk? Clarification is needed.
  3. There is a serious lack of information about the scanner parameters. In particular, no information are reported about how the MASH phantom images were virtually generated? Did you simulated a scanner? Which scanner parameters have been used? Why?
  4. On the previous point topic, the author decided to utilize specific scanner parameters to scan the 5-years old phantom. Why? I am sure that there is a reason, but it needs to be clarified to help the readers and potential researchers that would like to replicate the study.
  5. It seems like a fixed current technique was utilized. Why such choice? Only few exams are performed with fixed current in modern radiology. There is a need to clarify the potential impact of tube current modulation that can change the traditional statistical relationships between radiation dose, image quality, and patient size.
    1. Ria et. al, Technical note: Validation of TG 233 phantom methodology to characterize noise and dose in patient CT data, 2020, Medical Physics 47(4), https://doi.org/10.1002/mp.14089)
    2. Ria, J. T. Davis, J. B. Solomon, J. M Wilson, T. B. Smith, D. P. Frush, E. Samei. Expanding the concept of Diagnostic Reference Levels to Noise and Dose Reference Levels in CT. AJR. 2019; 213:889-894.
  6. The study, like every scientific study, has several limitations. Such limitations need to be acknowledged and reported in the paper (usually in the discussion).

Minor comments

  1. It is not clear in which slice and/or anatomical region where placed the ROIs used to calculate signal, noise, COV, etc. One more time, it is crucial to report as much information as possible about the utilized methodology as well as the reason behind such choice.
  2. There is not any discussion about different reconstruction algorithms (FBP or Iterative reconstruction, for instance) can affect the results reported in the study.

Author Response

(The authors gave the same response as above.)

Round 2

Reviewer 1 Report

I think the paper is now acceptable for publication.

Author Response

Thank you for your consideration.

Reviewer 2 Report

Thank you for proposing a new version of the manuscript.

Despite the proposed manuscript improved the previous version, I do not think that the majority of my concerns have been addressed.

In particular, adding a reference does not mean that a concern has been addressed. For instance, when I asked, in major comment number 1, why the authors decided to follow the reported methodology, I am asking for a reason, for a motivation why such methodology has been followed. The references can support such choice, but the authors should clearly report why they believe that their methodology is robust, and therefore deserves to be published.

In detail:

Previous letter major comment number 1. Please address my concern with adequate discussion in the manuscript.

Previous letter major comment number 2 and 4. The choice of utilize an adult and a pediatric phantom is not justified in the manuscript. Please, address this issue. Also, please justify why you decided to use MASH and ATOM phantoms.

Previous letter major comment number 5. You explained to me in the letter why you did not use modulated tube current techniques, but this should be also explained to the readers. Please change the manuscript accordingly.

Previous letter minor comment number 2. Again, adding a reference is not sufficient to address a concern. Please reflect your response also in the manuscript.

General comment: if it is not possible to address the reviewers' concerns, it is a good rule to at least acknowledge such limitations in the manuscript.

I would also recommend to clearly report in the point by point response letter every single modified sentence.

Thank you for the opportunity of reviewing this manuscript.

Author Response

(The authors gave the same response as above.)

Round 3

Reviewer 2 Report

I want to thank the authors for their kindness and for addressing all my comments.

I recommend the manuscript to be published.

Thank you.